# Fucoidan (*Undaria pinnatifida*)/Polydopamine Composite-Modified Surface Promotes Osteogenic Potential of Periodontal Ligament Stem Cells

**DOI:** 10.3390/md20030181

**Published:** 2022-02-28

**Authors:** Kyu Hwan Kwack, Ju Young Ji, Borami Park, Jung Sun Heo

**Affiliations:** 1Department of Oral Biology, School of Dental Medicine, University at Buffalo, State University of New York, New York, NY 14214, USA; hahahh@hanmail.net; 2Department of Maxillofacial Biomedical Engineering, Institute of Oral Biology, School of Dentistry, Kyung Hee University, 26 Kyunghee-daero, Dongdaemun-gu, Seoul 02447, Korea; wndud01099@naver.com (J.Y.J.); ami12010125@gmail.com (B.P.)

**Keywords:** fucoidan, polydopamine, periodontal ligament stem cells, osteogenic differentiation, transcriptome profiling

## Abstract

Fucoidan, a marine-sulfated polysaccharide derived from brown algae, has been recently spotlighted as a natural biomaterial for use in bone formation and regeneration. Current research explores the osteoinductive and osteoconductive properties of fucoidan-based composites for bone tissue engineering applications. The utility of fucoidan in a bone tissue regeneration environment necessitates a better understanding of how fucoidan regulates osteogenic processes at the molecular level. Therefore, this study designed a fucoidan and polydopamine (PDA) composite-based film for use in a culture platform for periodontal ligament stem cells (PDLSCs) and explored the prominent molecular pathways induced during osteogenic differentiation of PDLSCs through transcriptome profiling. Characterization of the fucoidan/PDA-coated culture polystyrene surface was assessed by scanning electron microscopy and X-ray photoelectron spectroscopy. The osteogenic differentiation of the PDLSCs cultured on the fucoidan/PDA composite was examined through alkaline phosphatase activity, intracellular calcium levels, matrix mineralization assay, and analysis of the mRNA and protein expression of osteogenic markers. RNA sequencing was performed to identify significantly enriched and associated molecular networks. The culture of PDLSCs on the fucoidan/PDA composite demonstrated higher osteogenic potency than that on the control surface. Differentially expressed genes (DEGs) (*n* = 348) were identified during fucoidan/PDA-induced osteogenic differentiation by RNA sequencing. The signaling pathways enriched in the DEGs include regulation of the actin cytoskeleton and Ras-related protein 1 and phosphatidylinositol signaling. These pathways represent cell adhesion and cytoskeleton organization functions that are significantly involved in the osteogenic process. These results suggest that a fucoidan/PDA composite promotes the osteogenic potential of PDLSCs by activation of critical molecular pathways.

## 1. Introduction

A variety of biomaterials, such as ceramics, polymers, and nanomaterials, are used to modify the surface properties of bone tissue-implantable devices [1,2,3]. An ideal coating substrate for implantable material must be both biocompatible and implantable and support the recruitment, adhesion, proliferation, and differentiation of cells both in vitro and in vivo. For bone tissue engineering applications, promising osteoconductive biomaterials must recreate a biophysical microenvironment similar to the extracellular matrix (ECM), which regulates the behavior and function of osteogenic-related cells. Thus, the discovery and development of advanced biomaterials that mimic the ECM is a crucial issue in bone regeneration and repair medicine.

The fabrication of surfaces with multiple biological functions has employed polymeric substances, such as proteins [4,5] and polysaccharides [6,7], for biomimetic ECM construction. Many natural polysaccharides have hydrogel properties that can potentially serve as an ECM within moist circumstances [8,9]. One of these natural polysaccharides, fucoidan, a marine-sulfated polysaccharide from the cell wall of brown algae, has been studied due to its anticoagulant [10], antifouling [11], and anti-inflammatory properties [12]. These biological actions of fucoidan can be influenced by its chemical construction with different sulfate and monosaccharide compositions. For instance, a main difference between fucoidan derived from *Undaria pinnatifida* and that from other brown algae is the monosaccharide formation of polysaccharide molecules [13,14]. Recently, fucoidan has been introduced as a suitable biomaterial for bone tissue regeneration [15,16,17]. Fucoidan from *Undaria pinnatifida* has been reported to enhance the osteogenic differentiation of human adipose-derived stem cells [18]. Other research suggests that fucoidan from *Undaria pinnatifida*-supplemented biocomposites can be a potential biomaterial for bone tissue regeneration [19]. Similar to fucoidan from *Undaria pinnatifida*, MSCs cultured on fucoidan from a *Fucus Vesiculosus*-containing composite scaffold showed increased mineral deposition [17,20]. Additionally, the functionalized scaffold composed of fucoidan (from *Fucus Vesiculosus*), a bioactivator in stem cell differentiation and nano-hydroxyapatite (nHA), is an inducing factor in bone mineralization attributed to bone tissue engineering applications [21]. Although the chemical composition of fucoidan differs between brown algae species, the bioactivity of fucoidan can support an osteogenic microenvironment that may ultimately reconstruct bone tissue. However, the precise molecular mechanisms underlying the observed effects of fucoidan within these substrates remain unknown.

Marine-derived polysaccharides have attracted high attention to fabricating functional matrices for biomedical devices and implants. However, these biomaterials cannot accomplish vigorous mechanical properties for hydrogelling [22,23]. Therefore, it is necessary to study which physicochemical conditions can modify and reinforce the rheological properties of natural polysaccharides. Previous studies demonstrated that the introduction of gelatin, a protein derived from collagen, into the alginate, a polysaccharide found in *Fucus vesiculosus*, increased the yield stress and viscosity of the hydrogels [23]. Corresponding with these issues, the present study employed mussel-inspired polydopamine (PDA), containing catechol and amine groups, to affix fucoidan to surfaces efficiently. Given the strong adhesive property and chemical stability of PDA, it has been utilized for the functional hydrogelling of polysaccharides to coat various substrates [24,25]. A recent study reported that the addition of PDA strengthened hemicellulose hydrogels with high mechanical properties [26]. Moreover, the incorporation of PDA into the dextran hydrogel enhanced the adhesion and drug loading capacity [27]. From a biological point of view, a heparin/PDA-conjugated film provided an effective feeder-free culture system for pluripotent human embryonic stem cells [28]. Data suggest that polysaccharide/PDA coatings can impact osteogenesis. A composite of a natural amino-polysaccharide, carboxymethyl chitosan, and PDA enhanced the osteogenic potential of human pluripotent stem cells in vitro [29]. Furthermore, a chondroitin sulfate/PDA-modified graft strengthened the osteogenic differentiation of MSCs and promoted new bone formation in vivo [30].

Periodontal ligament (PDL), a specialized tissue, anchors tooth to the bone socket in the jaw and manages tooth homeostasis, immune response, and repair [31]. Periodontal ligament stem cells (PDLSCs) are a mesenchymal stem cell (MSCs) population derived from PDL tissue and can differentiate into various types of mesenchymal-lineage cells, including periodontal tissue cells and bone-forming osteoblasts under proper experimental conditions [32,33,34]. A previous clinical pilot study demonstrated the potential competence and safety of autologous PDLSCs in the treatment of periodontal osseous defects [35]. Interestingly, human PDLSCs presented much higher growth capacity than human bone marrow-derived MSCs [36]. In terms of the ease of access of PDLSCs compared to other types of stem cells, PDLSCs are currently suggested as a useful source for stem cell-based approaches for regeneration of not only periodontal tissues but also bone tissues.

The present study herein designed a fucoidan and PDA composite-based hydrogel film for use in culture platforms to test if fucoidan could provide the artificial microenvironments necessary to intensify osteogenic potential using a periodontal ligament stem cells (PDLSCs) model. Additionally, we classify the critical molecular signaling pathways arising from fucoidan/PDA substrate that influence the osteogenic differentiation of PDLSCs. The findings suggest that a fucoidan/PDA composite is a useful strategy for the bio-coating of implantable devices used for bone tissue healing and regeneration.

## 2. Results

### 2.1. Fucoidan/PDA-Coated Surface Encourages the Osteogenic Differentiation of PDLSCs

The microstructure morphology of fucoidan/PDA composites was analyzed using SEM images. Distinct from the control surface, the fucoidan/PDA composites displayed a multilayered structure (Figure 1A). The fucoidan/PDA composites were also characterized by XPS to evaluate the surface chemistry. With the functionalization of fucoidan/PDA, carbon amounts decreased while amounts of oxygen, nitrogen, and particularly sulfur from the sulfate groups of fucoidan increased (Figure 1B). Given these results, the fucoidan/PDA composite successfully coated the PT surface.

Subsequently, the osteogenic differentiation of PDLSCs cultured on control and fucoidan/PDA-coated surfaces was evaluated by analyzing ALP activity, intracellular calcium levels ([Ca^2+^]_i_), and extracellular calcium deposits. ALP activity within cells on the fucoidan/PDA substrate increased compared to the control group on day 4, with a further increase on day 7 (Figure 1C). Increased [Ca^2+^]_i_ was also observed in cells on fucoidan/PDA substrates in a fucoidan dose-dependent manner (Figure 1D). Furthermore, extracellular calcium deposits were present with corresponding increases in PDLSCs on the fucoidan/PDA substrate compared with the control groups (Figure 1E).

To further determine the osteogenic effect of the fucoidan/PDA substrate on PDLSCs, the expression of osteogenic markers was assessed at the gene and protein level. The expression of osteogenic target genes, such as runt-related transcription factor 2 (RUNX2), osteopontin (OPN), collagen type I (COLI), osterix (OSX), and osteocalcin (OCN), was increased in cells on a fucoidan/PDA coating compared with the control group. The highest expression of osteogenic target genes was present in cells on 1 µg/mL of fucoidan/PDA (Figure 2A–E). In addition, OCN and RUNX2 protein levels were elevated in cells cultured on the fucoidan/PDA substrate on day 7 of osteogenic induction (Figure 2F). Taken together, these results suggest that the fucoidan/PDA substrate enhances the osteogenic potential of PDLSCs.

### 2.2. Identification of Differentially Expressed Genes (DEGs)

RNA sequencing was performed with PDLSCs cultured on the fucoidan/PDA substrate for 7 days. Genes displaying a fold-change cut-off of >2 in expression and a *p*-value < 0.05 were selected as significantly upregulated or downregulated genes in fucoidan/PDA substrate-cultured cells compared to the control group. Using these criteria, 348 DEGs (58 upregulated and 290 downregulated) were identified during fucoidan/PDA substrate-induced osteogenic induction (Appendix A). Cluster analysis of DEGs showed significant differences between cells cultured on the fucoidan/PDA substrate compared to the control substrate (Figure 3A). To determine the functions of the DEGs, gene ontology (GO) analysis was performed based on biological process and molecular function categories. The top 10 enriched GO terms were ranked. Briefly, Ras protein signal transduction and the phosphatidylinositol-3-phosphate biosynthetic process were significantly enriched in the cells cultured on the fucoidan/PDA substrate (Figure 3B). The GO terms related to molecular function showed that the DEGs were enriched in protein binding and DNA binding (Figure 3C). Validation of RNA-Seq data was performed by real-time RT-PCR with selected genes (EPCAM, NECAB3, HMOX1, PRPF38B, TRIM33, CAMSAP2). These genes were up or downregulated in a similar pattern of RNA-Seq analysis (Figure 3D).

### 2.3. KEGG Pathway Analysis

To further identify the biologically relevant signaling pathways of DEGs, KEGG pathway enrichment analysis was performed. From the KEGG pathway study, 10 pathways related to the DEGs were identified (Table 1).

Among these pathways, the DEGs were highly associated with biological pathways critical for osteogenic differentiation, including regulation of actin cytoskeleton, Ras-related protein 1 (Rap1) signaling, and phosphatidylinositol signaling (Figure 4, Figure 5 and Figure 6).

### 2.4. Protein–Protein Interaction (PPI) Network

Lastly, potential DEG interactions at the protein level were profiled using the STRING database to predict PPI networks. A total of 348 potential genes were uploaded to the STRING database to construct potential protein networks of DEGs (Figure 7).

The top hub of proteins with the highest number of PPI showed 11–21 interactions for each individual hub (Figure 8A). The top hubs of proteins include KRAS proto-oncogene, GTPase (KRAS), ATM serine/threonine kinase (ATM), phosphatidylinositol-4,5-bisphosphate 3-kinase catalytic subunit alpha (PIK3CA), RNA-binding motif protein 25 (RBM25), MutS homolog 2 (MSH2), and senataxin (SETX). Interestingly, the KRAS and ATM hub proteins had the highest number of interactions, with 21 proteins partners according to the STRING database (Figure 8B).

## 3. Discussion

This study finds that the polymeric combination of fucoidan/PDA composites can be an alternative design for biomimetic ECM construction to encourage osteogenic differentiation of PDLSCs. Current studies have shown the potential application of a natural biomaterial, fucoidan, in surface functionalization technology. Fucoidan has been shown to have characteristics useful for antifouling, cancer therapy, and vascular and bone regeneration [11,17,37]. Regarding osteogenic regeneration and bone health, fucoidan facilitates osteogenic-related cell proliferation through increased ALP activity, calcium deposition, and bone morphogenetic protein 2 (BMP2) levels, and it is connected with bone mineralization and development [38,39,40]. Fucoidan-based ceramic or protein-combined scaffolds accelerate osteogenesis and mineralization, which makes it a very promising composite technique in bone tissue engineering [16,41].

Unique from other reports of fucoidan-based composites as bone forming-biomaterials, the present study profiles the changing cellular transcriptome to further unveil molecular and functional pathways encouraged by fucoidan/PDA composites during osteogenic induction. KEGG pathway enrichment analysis revealed that significantly enriched DEGs include the regulation of actin cytoskeleton, the Rap1 signaling pathway, and the phosphatidylinositol signaling system. Effects of the actin cytoskeleton suggest a connection between the cell and the outside environment, which triggers cell-ECM interaction through various intracellular signaling networks [42,43]. The cytoskeleton network may serve to strengthen the osteogenic potential of PDLSCs in response to the fucoidan/PDA composite. Similar studies have reported actin cytoskeleton dynamics are commonly observed when MSCs and other osteogenic cells commit to osteogenic lineages [44,45,46,47]. A better understanding of the cytoskeletal network may provide notable insights into the osteogenic regulation of PDLSCs by fucoidan/PDA.

Interestingly, Rap1, a member of the Ras superfamily of small GTPase, is involved in cell-ECM adhesion and cell–cell junctions through reorganization of the actin cytoskeleton in mammalian cells [48,49]. Several studies have reported that Rap1 signaling promotes the osteogenic differentiation of MSCs and osteoblastic cells, although few data exist that show a role for Rap1 in regulation of osteogenic cell differentiation [50,51,52]. Rap1 may be a crucial marker for the complex network of cell-interface events involved in stem cell differentiation into the osteogenic lineage. As such, the fucoidan/PDA composite performs as an ideal ECM architecture mimic by crucially stimulating the Rap1-actin cytoskeleton network involved in cell adhesion to encourage the osteogenic differentiation of PDLSCs.

KEGG pathway analysis indicated that phosphatidylinositol signaling is a prominent pathway in PDLSCs plated on the fucoidan/PDA composite. Phosphatidylinositol signaling is involved in virtually all cellular functions, including cell proliferation, differentiation, and metabolic processes [53]. Phosphatidylinositol is classified as a glycerophospholipid that anchors to the cytoplasmic face of the cellular membrane. The phosphorylation of phosphatidylinositol leads to activation of the PI3K/Akt pathway, which includes key intracellular molecules in the phosphatidylinositol system involved in principle roles across multiple cellular processes [54]. Furthermore, phosphatidylinositol signaling influences cell adhesion and actin cytoskeleton assembly [55,56] and influences the differentiation of stem cells and osteoblasts and also bone homeostasis [57,58,59]. These data suggest that the actin cytoskeleton network, Rap1 signaling, and phosphatidylinositol signaling may be important mechanisms by which the ECM-like architecture of fucoidan/PDA promotes the osteogenic differentiation of PDLSCs.

Using RNA-Seq results to uncover PPI networks, we found several significant hub proteins. Consistent with KEGG analysis, cell adhesion and cytoskeleton assembly proteins, such as KRAS, a small GTPase, and phosphatidylinositol signaling, were identified as significant hubs. Additionally, the ATM serine/threonine kinase was found as a hub and is involved in focal adhesion and actin cytoskeleton rearrangement in cancer progression [60]. Another hub protein, senataxin, is an RNA/DNA helicase reported to regulate actin cytoskeleton organization for vesicle transport and autophagy regulation [61]. Taken together, this study suggests that the cell adhesion and cytoskeleton assembly events of a fucoidan/PDA substrate impact the differentiation of PDLSCs into osteogenic lineages.

Indeed, the biological activity of fucoidan is considerably influenced by different molecular composition, sulfates position, and the polysaccharide framework. Although it is necessary to perform chemical structure-dependent studies to elucidate the physicochemical properties of fucoidan as a functional ECM biomaterial, regarding the main focus of this study, we engineered a natural polysaccharide, fucoidan (*Undaria pinnatifida*), and PDA composite for ECM construction to enable PDLSC adhesion and osteogenic differentiation. Additionally, a comprehensive bioinformatic analysis was performed to elucidate the molecular mechanisms underlying fucoidan/PDA-induced PDLSC-osteogenic differentiation. Finally, these findings provide novel insights into molecular targets for the development of implantable devices for periodontal tissue and bone regeneration.

## 4. Materials and Methods

### 4.1. Materials

Fetal bovine serum (FBS) and α-MEM were obtained from Gibco-BRL (Gaithersburg, MD, USA). Chemicals, including fucoidan (from *Undaria pinnatifida*, F8315, ≥95% Purity) and dopamine hydrochloride, and laboratory wares were purchased from the Sigma-Aldrich (St. Louis, MO, USA) and SPL Lifescience (Pocheon, Korea), respectively. Antibodies were supplied by Santa Cruz Biotechnology (Santa Cruz, CA, USA).

### 4.2. Periodontal Ligament Stem Cell Culture

Human periodontal ligament stem cells were obtained from CELPROGEN (CELPROGEN, Torrance, CA, USA). The culture of PDLSCs was previously described [62]. Cells were cultured on cell culture polystyrene (PT) (control), PDA-coated, or fucoidan/PDA-coated surfaces with an osteogenic medium (α-MEM containing 5% FBS, 50 μg/mL ascorbic acid, 1 μM dexamethasone, and 3 mM β-glycerophosphate).

### 4.3. Preparation of the Fucoidan/PDA Composite Substrate

The PDA solution was prepared by dissolving 1 mg of L-DOPA in 1 mL of 10 mM Tris buffer base (pH 8.5; Sigma-Aldrich). Fucoidan was added to the PDA solution (1 mg/mL), and the mixture was stirred at room temperature for 24 h. The final concentration of fucoidan in the PDA solution was 0.5 or 1 µg/mL for each experiment. The PT culture surface was layered with a fucoidan/PDA composite solution overnight at room temperature and washed three times with sterile phosphate-buffered saline (PBS). Fucoidan/PDA-coated surfaces were then dried in a vacuum oven.

### 4.4. Characterization of the Fucoidan/PDA-Coated Surface

Each specimen was rinsed with PBS and freeze-dried overnight. Then the surface images of the fucoidan/PDA composite were assessed using scanning electron microscopy (SEM; S-4700, Hitachi, Schaumburg, IL, USA). X-ray photoelectron spectroscopy (XPS) was performed to evaluate the atomic composition of the fucoidan/PDA-modified surfaces as described in our previous study [62,63]. The surface composition concentration was acquired from the XPS survey spectra.

### 4.5. Alkaline Phosphatase (ALP) Activity

ALP activity was evaluated as described in our previous study [62,63]. Briefly, cells were lysed, total protein concentration was determined, 200 μL of p-nitrophenylphosphate (pNPP) were added to the cell lysate, and the mixture was incubated for 30 min at 37 °C. The reaction was then stopped by adding of 3 M NaOH. The standard curve was prepared using a p-nitrophenol solution (10 mM, Sigma-Aldrich), and each of the standards with p-nitrophenol concentrations was formulated between 0.2 and 100 μM. Absorbance of each sample and standard was measured with a microplate spectrophotometer (MicroDigital Co., Ltd., Seongnam, Korea) at 405 nm. The mean absorbance of the blank standard was subtracted from all standard and sample readings. The ALP activity was calculated from the linear regression equation obtained from the standard curve using Microsoft Excel. ALP activity was presented as μM/100 μg of protein.

### 4.6. Intracellular Calcium Quantification Assay

Intracellular calcium content was determined as described in our previous report [62,63]. Briefly, cells were cultured onto fucoidan/PDA composite-coated surfaces. Seven and fourteen days after osteogenic induction, the intracellular calcium level was quantified using a calcium assay kit (BioAssay Systems, Hayward, CA, USA) according to the manufacturer’s instructions. Standard solution (0~6 mg/dl) was prepared and read optical density with a microplate spectrophotometer at 612 nm. The absorbance of blank standard was subtracted from all standard and sample absorbance values. Calcium concentration of the sample was calculated from the linear regression equation obtained from standard curve. Calcium levels were converted as mg/100 mg of protein.

### 4.7. Alizarin Red S Staining

Alizarin Red S staining was performed to identify calcium deposits in cell cultures as described in our previous report [62,63]. Cells were fixed with 4% paraformaldehyde for 15 min after a 14-day osteogenic incubation. Cells were then rinsed three times with PBS and stained with 2% alizarin red S solution (pH 4.2) for 5 min. Stained cells were rinsed three times and observed with a light microscope (Olympus DP72, Tokyo, Japan).

### 4.8. RNA Extraction and Real-Time Reverse Transcriptase–Polymerase Chain Reaction (RT-PCR)

RNA extraction, cDNA synthesis, and real-time RT-PCR procedures were performed as previously described using a QuantiTect SYBR Green PCR kit (Qiagen) and icycler iQ Multi-color Real-time RT-PCR System [27]. The reaction conditions were 95 °C for 30 s, 95 °C for 5 s, 55 °C for 30 s, and 72 °C for 30 s for 40 cycles. The primers for osteogenic-related genes were as follows: 5′- GTCTCACTGCCTCTCACTTG-3′ (sense) and 5′-CACACATCTCCTCCCTTCTG-3′ (antisense) for RUNX2; 5′-GCAGACCTGACATCCAGTACC-3′ (sense) and 5′-GATGGCCTTGTATGCACCTTC-3′ (antisense) for OPN; 5′-GCGGCTCCCCATTTTTATACC-3′ (sense) and 5′-GCTCTCCTCCCATGTTAAATAGCAC-3′ (antisense) for COL I; 5′- ATGAGAGCCCTCACACTCTCG-3′ (sense) and 5′- GTCAGCCAACTCGTCACAGTCC-3′ (antisense) for OCN; 5′-TGAGGAGGAAGTTCACTATGG-3′ (sense) and 5′-TTCTTTGTGCCTGCTTTGC-3′ (antisense) for OSX; 5′-GCCAGTGTACTTCAGTTGGTGC-3′ (sense) and 5′-CCCTTCAGGTTTTGCTCTTCTCC (antisense) for EPCAM; 5′-GCATTGGAATCGCTGAACCGTG-3′ (sense) and 5′-CGCGTCACAAACTGGTCCACTT-3′ (antisense) for NECAB3; 5′-CCAGGCAGAGAATGCTGAGTTC-3′ (sense) and 5′- AAGACTGGGCTCTCCTTGTTGC-3′ (antisense) for HMOX1; 5′-CCAACATCCTGTCGTCGCCTTA-3′ (sense) and 5′-CCCGCTGTTTTCCTGCTTCCTT-3′ (antisense) for PRPF38B; 5′-CCTCAGTTACCAATCCAGAAAACC-3′ (sense) and 5′- GGTAGGTGACTGCCTGAGATGT-3′ (antisense) for TRIM33; 5′-GGAGGTCAAAAGGCTCGTTATCG-3′ (sense) and 5′- GGCAGCTAATGCACAGCCATCT-3′ (antisense) for CAMSAP2; and 5′- GCTCTCCAGAACATCATCC-3′ (sense) and 5′-TGCTTCACCACCTTCTTG-3′ (antisense) for GAPDH.

### 4.9. Western Blot Analysis

Total protein extraction and SDS-polyacrylamide gel electrophoresis were performed as previously described [34]. Primary (anti-OCN, anti-RUNX2, or anti-β-actin; Santa Cruz Biotechnology) and secondary antibodies (goat anti-rabbit immunoglobulin G (IgG) or goat anti-mouse IgG conjugated to horseradish peroxidase) were used to identify the osteogenic differentiation of PDLSCs.

### 4.10. Library Preparation and Sequencing

RNA sequencing was performed as described previously [64]. In brief, total RNA was extracted, and the library was constructed using a QuantSeq 3′ mRNA-Seq Library Prep Kit (Lexogen, Inc., Vienna, Austria) according to the manufacturer’s instructions. The library was amplified to add the complete adapter sequences required for cluster production. The completed library was purified from PCR components, and high-throughput sequencing was then achieved as single-end 75 sequencing using NextSeq 500 (Illumina, Inc., San Diego, CA, USA).

### 4.11. Data Analysis

QuantSeq 3′ mRNA-Seq reads were aligned using Bowtie2 [65]. Bowtie2 indices were generated from either the genome assembly sequence or the representative transcript sequences for aligning to the genome and transcriptome. The alignment file was applied for assembling transcripts and assessing the amounts and differential expression. Differentially expressed genes (DEGs) were evaluated based on counts from unique and multiple alignments using coverage in Bedtools [66]. The RC (Read Count) data were acquired from the quantile normalization method using EdgeR within R by Bioconductor [67]. Gene classification and ontology were searched using DAVID (http://david.abcc.ncifcrf.gov/, accessed on 27 August 2021.) and Medline databases (http://www.ncbi.nlm.nih.gov/, accessed on 27 August 2021). Data mining and graphic visualization were performed using ExDEGA (Ebiogen Inc., Seoul, Korea). The construction of the target protein–protein interaction (PPI) network was performed for protein targets using STRING v3.8.2. A PPI score of >0.4 (*p* < 0.05) was considered significant. The PPI networks were visualized using Cytoscape software (http://www.cytoscape.org, accessed on 15 September 2021).

### 4.12. Statistical Analysis

Data are expressed as means ± standard deviation using SPSS software (ver. 10.0; SPSS Inc., Chicago, IL, USA). One-way analysis of variance was applied for multiple comparisons (Duncan’s multiple range test). A *p* value < 0.05 was considered a statistically significant difference.

## Figures and Tables

**Figure 1 marinedrugs-20-00181-f001:**
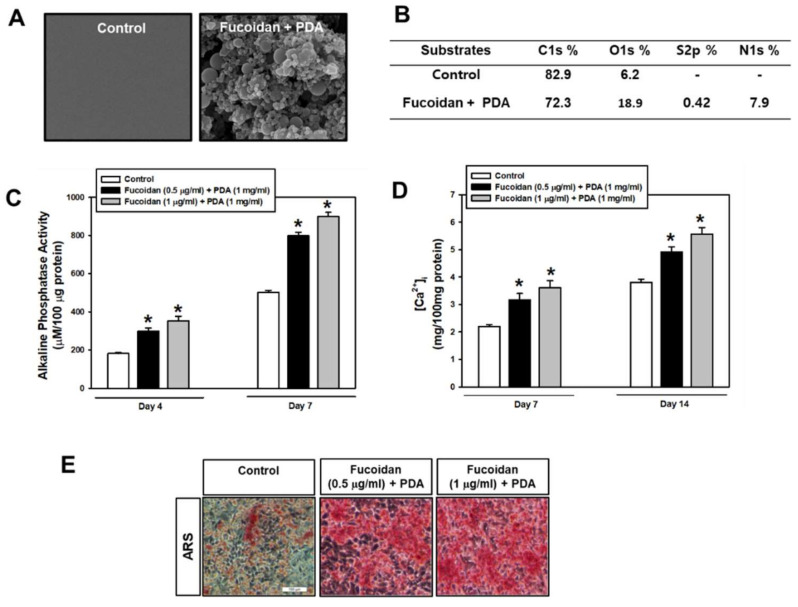
Surface morphology of fucoidan/PDA composite. (**A**) Scanning electron microscopy images of the PT surface (control) and fucoidan/PDA (1 µg/mL + 1 mg/mL)-coated surfaces at ×10 magnification. (**B**) Relative atomic composition of each surface. Cells were cultured on the fucoidan/PDA composite (PDA with 1 mg/mL and variable fucoidan concentration with 0.5 or 1 µg/mL) for 4, 7, and 14 days. (**C**) ALP activity, (**D**) [Ca^2+^]_i_ and (**E**) Alizarin Red S staining were assessed (magnification 200×). Values are presented as means ± SD (*n* = 4, * *p* < 0.05 vs. the control value at each time point).

**Figure 2 marinedrugs-20-00181-f002:**
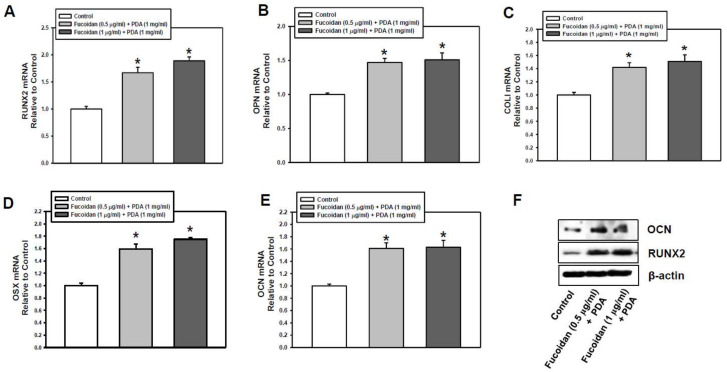
Effect of fucoidan/PDA composite on osteogenic markers. (**A**–**E**) The mRNA expression of RUNX2, OPN, COLI, OSX, and OCN was evaluated by real-time RT-PCR after 7-day osteogenic induction. (**F**) Protein levels of OCN (5.5 kDa) and RUNX2 (55 kDa) were analyzed by Western blot analysis. The values are denoted as means ± SD (*n* = 3, * *p* < 0.05).

**Figure 3 marinedrugs-20-00181-f003:**
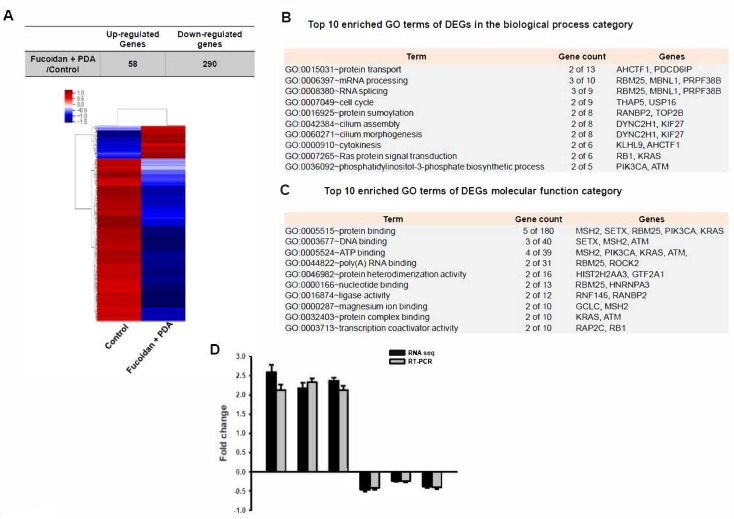
RNA sequencing analysis of PDLSCs cultured on the fucoidan/PDA composite. (**A**) Numbers of upregulated and downregulated genes in cells on fucoidan/PDA compared to those in control groups with at least a twofold change in expression level. Hierarchical clustering of significant DEGs. Red represents upregulation; blue represents downregulation. The top 10 enriched gene ontology terms in the (**B**) biological process and (**C**) molecular function categories from DAVID gene ontology term analysis for DEGs. (**D**) The expressions of three up and three down genes (randomly selected) were evaluated by real-time RT-PCR to validate the RNA-Seq data. The values are presented as means ± SD of three independent experiments.

**Figure 4 marinedrugs-20-00181-f004:**
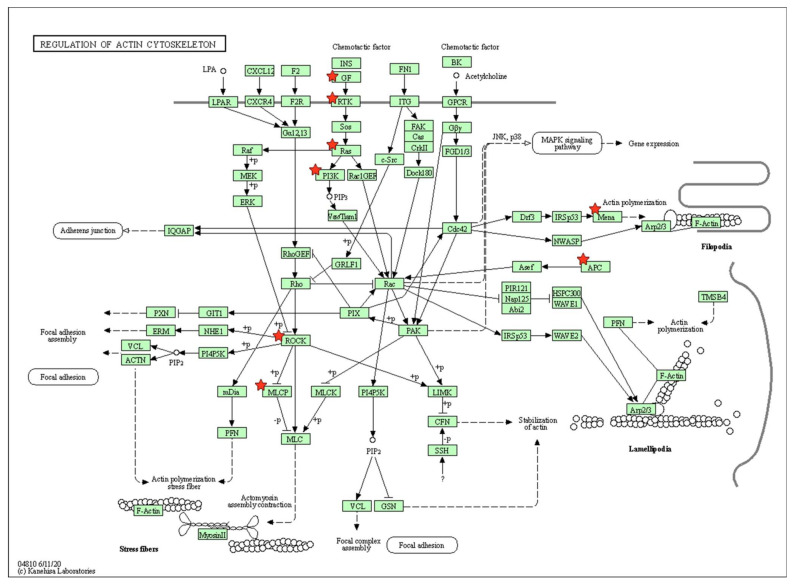
KEGG pathway analysis: regulation of actin cytoskeleton. The red stars represent key player genes.

**Figure 5 marinedrugs-20-00181-f005:**
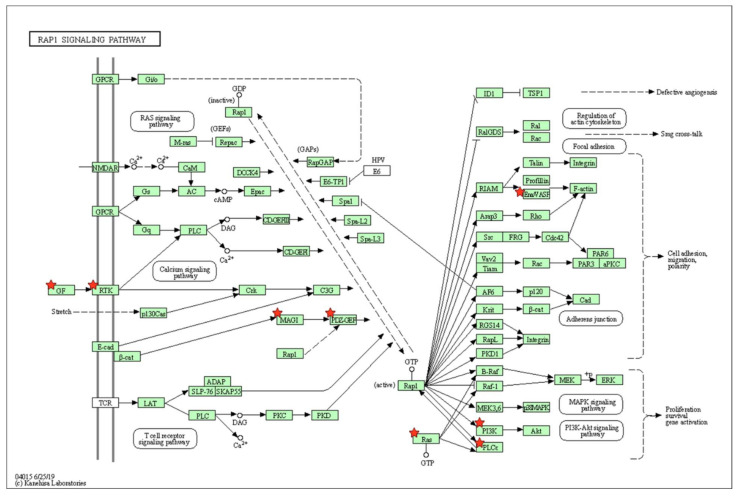
KEGG pathway analysis: Rap1 signaling pathway.

**Figure 6 marinedrugs-20-00181-f006:**
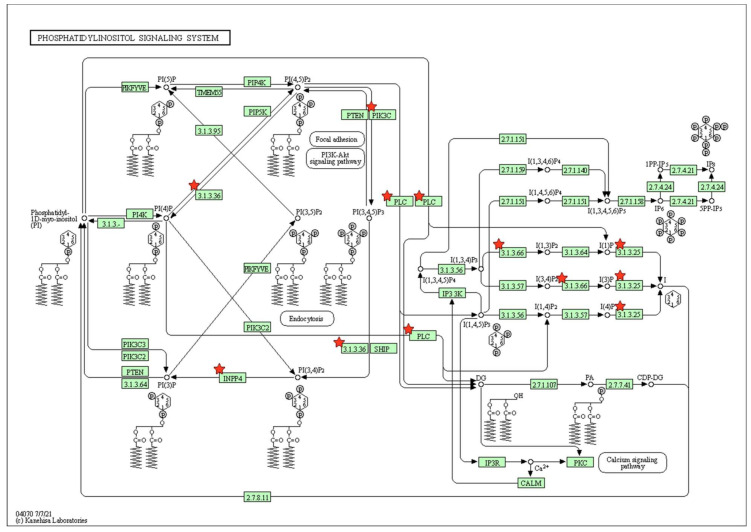
KEGG pathway analysis: phosphatidylinositol signaling system.

**Figure 7 marinedrugs-20-00181-f007:**
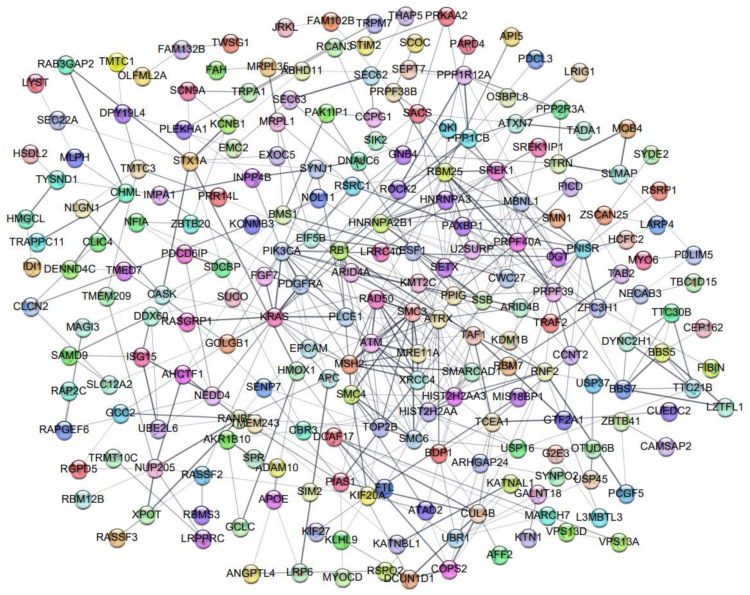
The protein–protein interaction network of DEGs was analyzed based on the STRING database. Each node depicts a protein. The thickness of lines was based on the strength of data support.

**Figure 8 marinedrugs-20-00181-f008:**
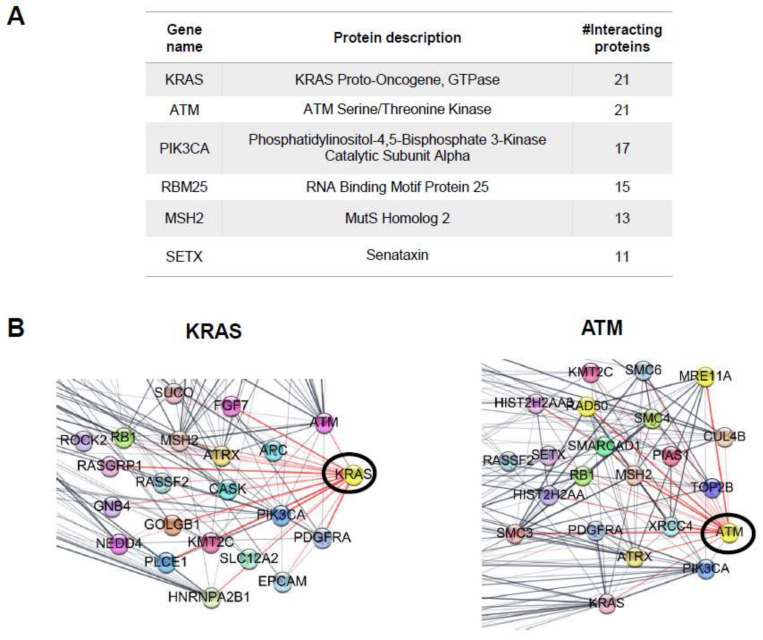
Top hubs of networks. (**A**) Top hub proteins with the highest number of interactions. (**B**) KRAS and ATM hub networks. KRAS and ATM represent hub proteins with the highest number of predicted protein interactions, numbering 21.

**Table 1 marinedrugs-20-00181-t001:** Top 10 enriched KEGG pathways analysis. Potential genes (*n* = 348) were imported into DAVID tools. Categories are based on the KEGG database.

Category	Term	Count	Genes
KEGG_PATHWAY	Pathways in cancer	11	RB1, PDGFRA, FGF7, APC, MSH1, PIK3CA, ROCK2, GNB4, TRAF2, KRAS
KEGG_PATHWAY	Regulation of actin cytoskeleton	9	APC, KRAS, ROCK2, ENAH, FGF7, PIK3CA, PDGFRA, PPP1CB, PPP1R12A
KEGG_PATHWAY	Rap1 signaling pathway	7	KRAS, RAPGEF6, FGF7, MAGI3, PIK3CA, PLCE1, PDGFRA
KEGG_PATHWAY	Oxytocin signaling pathway	6	PPP1CB, OXTR, PPP1R12A, ROCK2, KRAS
KEGG_PATHWAY	Inositol phosphate metabolism	5	INPP4B, SYNJ1, PIK3CA, IMPA1, PLCE1
KEGG_PATHWAY	Melanoma	5	RB1, PDGFRA, FGF7, PIKC3A, KRAS
KEGG_PATHWAY	Phosphatidylinositol signaling system	5	INPP4B, SYNJ1, PIK3CA, IMPA1, PLCE1
KEGG_PATHWAY	Mineral absorption	4	HMOX1, TRPM7, CLCN2, FTL
KEGG_PATHWAY	Colorectal cancer	4	APC, MSH2, PIK3CA, KRAS
KEGG_PATHWAY	Glioma	4	RB1, PDGFRA, PIK3CA, KRAS

## Data Availability

Not applicable.

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
