# Peer review of "Fucoidan (Undaria pinnatifida)/Polydopamine Composite-Modified Surface Promotes Osteogenic Potential of Periodontal Ligament Stem Cells"

_marinedrugs, 2022, doi:10.3390/md20030181_

Round 1

Reviewer 1 Report

Dear authors!

I read the manuscript and I had some questions and recommendations.

  1. It is known that the origin, technology, molecular weight, purity and other factors have a key influence on the biological activity of fucoidan. Give a description of the fucoidan used in your work. Fucoidan from F.vesiculosus has been found to have concentration-dependent inhibition of hyaluronidase enzyme (https://doi.org/10.3390/md18050275). Discuss this aspect in terms of its effect on periodontal ligament stem cells.
  2. For ALP-activity methods, a model of spectrophotometer, the range of method linearity, and its accuracy are introduced.
  3. For intracellular calcium quantification, instrument model, linearity range, and accuracy are used.
  4. For Alizarin Red S staining, specify microscope model and magnification.
  5. It has been shown that polymers must be added to the fucoidan solution to impart the necessary rheological properties (https://doi.org/10.1016/j.fbio.2019.03.002). How did you evaluate the quality of your particles? Discuss this aspect of the need for modification or justify the absence of modifiers.
  6. As a result of the experiment, you do not have a dose-dependence for fucoidan. Based on what literature or experimental data did you choose 0.5 and 1.0 µg/ml concentration of fucoidan? 

Author Response

  1. It is known that the origin, technology, molecular weight, purity and other factors have a key influence on the biological activity of fucoidan. Give a description of the fucoidan used in your work. Fucoidan from F.vesiculosus has been found to have concentration-dependent inhibition of hyaluronidase enzyme (https://doi.org/10.3390/md18050275). Discuss this aspect in terms of its effect on periodontal ligament stem cells.

: We really appreciate your kind comments and advice to have a chance for improving the current research. As your important comments, we present that fucoidan in this research is originated from Undaria pinnatifida (Cat. F8315, Sigma-Aldrich). A brown seaweed, Undaria pinnatifida containing fucoidan is broadly cultivated in East Asia and it is a plentiful source from which fucoidan is commonly extracted. It has been known that a main difference between fucoidan from Undaria pinnatifida and those from other brown algae such as Fucus vesiculosus is monosaccharide formation of polysaccharide molecule (1, 2). It is also suggested that the biological action of fucoidan can be influenced by its chemical construction with different sulfate and monosaccharide composition. Previous study has shown that fucoidan from Undaria pinnatifida enhanced osteogenic differentiation of human adipose-derived stem cells (3). Other report suggests that fucoidan from Undaria pinnatifida-supplemented biocomposites may be a potential biomaterial for bone tissue regeneration (4). Based on these circumstances and issues, the present study employed fucoidan from Undaria pinnatifida and designed a fucoidan and polydopamine (PDA) composite-based film for use in a culture platform for periodontal ligament stem cells (PDLSCs) and explored the prominent molecular pathways induced during osteogenic differentiation of PDLSCs through transcriptome profiling. We stated this subject in the text (Introduction section). Thank you again for your hard working for reviewing our manuscript.

References:

(1) Vishchuk, O.S.; Ermakova, S.P.; Zvyagintseva, T.N. Sulfated polysaccharides from brown seaweeds Saccharina japonica and Undaria pinnatifida: Isolation, structural characteristics, and antitumor activity. Carbohydr. Res. 2011, 346, 2769–2776.

(2) Kalimuthu, S.; Kim, S.K. Fucoidan, A Sulfated Polysaccharides from Brown Algae as Therapeutic Target for Cancer; Springer International Publishing: Cham, Switzerland, 2015; pp. 145–164, ISBN 978-3-319-07144-2.

(3) Park SJ, Lee KW, Lim DS, Lee S. The sulfated polysaccharide fucoidan stimulates osteogenic differentiation of human adipose-derived stem cells. Stem Cells Dev. 2012 Aug 10;21(12):2204-11.

(4) Lee JS, Jin GH, Yeo MG, Jang CH, Lee H, Kim GH. Fabrication of electrospun biocomposites comprising polycaprolactone/fucoidan for tissue regeneration. Carbohydr Polym. 2012 Sep 1;90(1):181-8.

  1. For ALP-activity methods, a model of spectrophotometer, the range of method linearity, and its accuracy are introduced.

: As your important comments, we revised the procedure of ALP activity including a model name of spectrophotometer in the ‘Materials and Methods section’. We also added the linearity test (standard curve) for ALP activity calculation in supplementary information (Fig. S1). Absorbance of each sample displayed within the linear range of the standard curve (O.D. 0 ~1.37 of 0.2 ~100 µM p-nitrophenol). The mean absorbance of the blank standard was subtracted from all standard and sample readings. Moreover, we found a miswriting of ALP activity unit and corrected previous unit (mM/100 μg of protein) to (µM/100 μg of protein). We hope your kind understanding for this mistake.

  1. For intracellular calcium quantification, instrument model, linearity range, and accuracy are used.

: As your critical indication, we revised ‘intracellular calcium quantification’ in Methods section. We also presented the standard curve in supplementary information (Fig. S2). According to the manufacturer’s instructions, we prepared standard solution (0 ~ 6 mg/dl) and read optical density at 612 nm. The blank standard absorbance was subtracted from all standard and sample absorbance value. Calcium concentration of the sample is calculated from linear regression equation obtained from standard curve. Since calcium content was evaluated with total protein extracted from cell lysate, calcium levels were converted as mg/100 mg of protein based on our published researches (1, 2).

References:

(1) Shim, N.Y.; Ryu, J.; Heo, J.S. Osteoinductive function of fucoidan on periodontal ligament stem cells: Role of PI3K/Akt and Wnt/β‐catenin signaling pathways. Oral Dis. 2021, Epub.

(2) Shim, N.Y.; Heo, J.S. Performance of the Polydopamine-Graphene Oxide Composite Substrate in the Osteogenic Differenti-ation of Mouse Embryonic Stem Cells. Int. J. Mol. Sci. 2021, 22, 7323.

  1. For Alizarin Red S staining, specify microscope model and magnification.

: As your indication, we stated the microscope model (Olympus DP72, Japan) and magnification (200×) in the text (ARS method section and figure 1 legend).

  1. It has been shown that polymers must be added to the fucoidan solution to impart the necessary rheological properties (https://doi.org/10.1016/j.fbio.2019.03.002). How did you evaluate the quality of your particles? Discuss this aspect of the need for modification or justify the absence of modifiers.

: As your valuable comments, we can have a chance to study the specific rheological properties of various polysaccharides including fucoidan and other polymers. We appreciate for your advice. We stated this issue that you suggest in ‘Introduction’ section.

References:

(1) Bhutani, U., Laha, A., Mitra, K., & Majundar, S. Sodium alginate and gelatin hydrogels: Viscosity effect on hydrophobic drug release. Materials Letters, 2016; 164, 76–79.

(2) Voron'koa NG, Derkacha SR, Kuchinaa YA, Sokolana NI, Kuranovaa LK, Obluchinskaya ED. Influence of added gelatin on the rheological properties of a Fucus vesiculosus extract. Food Bioscience. 2019: 29: 1-8.

(3) Ge J, Gu K, Sun K, Wang X, Yao S, Mo X, Long S, Lan T, Qin C. Preparation and Swelling Behaviors of High-Strength Hemicellulose-g-Polydopamine Composite Hydrogels. Materials (Basel). 2021;14(1):186.

(4) Zhang M, Huang Y, Pan W, Tong X, Zeng Q, Su T, Qi X, Shen J. Polydopamine-incorporated dextran hydrogel drug carrier with tailorable structure for wound healing. Carbohydr Polym. 2021;253:117213.

  1. As a result of the experiment, you do not have a dose-dependence for fucoidan. Based on what literature or experimental data did you choose 0.5 and 1.0 µg/ml concentration of fucoidan?

: The selected 0.5 and 1.0 µg/ml concentration of fucoidan was based on our previous study (1). In previous study, we performed various molecular and cellular experiments to evaluate the osteogenic effect of fucoidan (0, 0.1, 0.5, 1, 2 µg/ml) on periodontal ligament stem cells (PDLSCs) and found the stable and significant osteogenic differentiation of PDLSCs with 0.5 and 1.0 µg/ml of fucoidan. According to these previous results, the present study employed two points of concentration for fucoidan treatment. Thank you again for your important indication.

References:

(1) Shim NY, Ryu JI, Heo JS. Osteoinductive function of fucoidan on periodontal ligament stem cells: Role of PI3K/Akt and Wnt/β-catenin signaling pathways. Oral Dis. 2021 Mar 7. doi: 10.1111/odi.13829.

Reviewer 2 Report

This is a very interesting article.

I suggest the authors to enlarge figure 8.

I would also suggest the authors to talk about nanohydroxiapatite. They can get informations from this article DOI 10.1080/03602532.2020.1758713.

The authors have also to mention the limitations of the present study.

Author Response

  1. I suggest the authors to enlarge figure 8.

: We really appreciate your kind comments and advice to have a chance for improving the current research. As your indication, we changed figure 8 to the enlarged version.

  1. I would also suggest the authors to talk about nanohydroxiapatite. They can get informations from this article DOI 10.1080/03602532.2020.1758713.

: As your suggestion, we discussed the issue of nanohydroxyapatite in the text (Introduction section). This improves the idea of the present study much wider to explain feasible biocompatible materials for dental tissue and bone regeneration.

References:

(1) Lowe B, Venkatesan J, Anil S, Shim MS, Kim SK. Preparation and characterization of chitosan-natural nanohydroxyapatite-fucoidan nanocomposites for bone tissue engineering. Int J Biol Macromol. 2016; 93:1479–1487.

  1. The authors have also to mention the limitations of the present study.

: As your important comments, we discussed the limitations of the present study in ‘Discussion’ section.

Round 2

Reviewer 1 Report

Dear Authors,

  1. Link 23 is not formatted correctly. Correct to "Voron'ko, N.G., Derkach, S.R., Kuchina, Y.A., Sokolan, N.I., Kuranova, L.K., Obluchinskaya, E.D...."
  2. In section 4.1, indicate the catalog number of fucoidan and its characteristics (purity, monosaccharide composition, etc.).
  3. Please correct the title of the manuscript according to the fucoidan used.
  4. Fucoidan is a heteromolecule of complex composition, the pharmacological and biological properties of which depend on the type of algae, the technology of isolation and purification, the monosaccharide composition, and so on. Adjust the conclusions of your manuscript with respect to your object. 

Author Response

  1. Link 23 is not formatted correctly. Correct to "Voron'ko, N.G., Derkach, S.R., Kuchina, Y.A., Sokolan, N.I., Kuranova, L.K., Obluchinskaya, E.D...."

: Thank you for your dedicated review working for our manuscript. As your indication, we checked the reference 23 format, however, we think the present listed format is correct for ‘Marine Drugs’. If any other comment you have, please let us know.  

  1. Voron'koa, N.G.; Derkacha, S.R.; Kuchinaa, Y.A.; Sokolana, N.I.; Kuranovaa, L.K.; Obluchinskaya, E.D. Influence of added gelatin on the rheological properties of a Fucus vesiculosus extract. Food Bioscience. 2019, 29, 1-8.

  1. In section 4.1, indicate the catalog number of fucoidan and its characteristics (purity, monosaccharide composition, etc.).

: As your important indication, we surveyed fucoidan characteristics via Sigma-Aldrich company homepage, but we cannot find any specific information of fucoidan composition. Thus, we provide the catalog number of fucoidan and its purity that the company officially announces (F8315, ≥ 95 % Purity). This also referred to a previous study to show fucoidan characteristics as ‘fucoidan (F8315, ≥ 95)’ (1).

Reference:

(1) Ponnan A, Kulanthaiyesu A, Marudhamuthu M, Palanisamy K, Kadarkarai M. Protective effects of fucoidan against 4-nitroquinolin-1-oxide provoked genetic damage in mouse bone marrow cells. Environ Sci Pollut Res Int. 2020 Sep;27(25):31760-31766.

  1. Please correct the title of the manuscript according to the fucoidan used.

: As your kind comments, we corrected the title of this manuscript to ‘Fucoidan (Undaria pinnatifida)/polydopamine composite-modified surface promotes osteogenic potential of periodontal ligament stem cells’

  1. Fucoidan is a heteromolecule of complex composition, the pharmacological and biological properties of which depend on the type of algae, the technology of isolation and purification, the monosaccharide composition, and so on. Adjust the conclusions of your manuscript with respect to your object.

: As your important indication, we adjusted the conclusion according to the object of the present manuscript.

Round 3

Reviewer 1 Report

I have no more questions.